# Recovery time and predictors of severe acute malnutrition in children aged 6–59 months via an outpatient therapeutic program in Borena zone: *A prospective cohort study*

**Girma Tenkolu Bune** [1] *, **Abuna Mohammed**[2], **Samrawit Hailu**[1], **Eden Ashenafi**[2]

1 School of Public Heath, College of Medical and Health Sciences, Dilla University, Dilla, Ethiopia,
2 Department of Reproductive Health, College of Medical and Health Sciences, Dilla University, Dilla, Ethiopia

* girmatbune@gmail.com

## Abstract

### Background

Severe acute malnutrition (SAM) is a severe condition causing bilateral pitting edema or signs of wasting in children, with a high mortality risk. An outpatient therapeutic program is recommended for managing SAM children without complications, but there is limited information on recovery time and its determinants.

### Objective

This study aims to assess the time to recovery and its predictors among children aged 6–59 months with SAM admitted to the Outpatient therapeutic program in the Borena zone, Oromia region, Southern Ethiopia in 2023.

### Methods

A prospective follow-up study was carried out from March 1–30, 2023, on 322 children aged 6–59 months in health facilities in the Borena zone. After being collected using a structured questionnaire, the data was imported into Epi Data Manager version 3.1 and exported to SPSS version 26 for analysis. Model fitness was assessed using the log rank test, and time-to-recovery from SAM was found to be predicted by Cox regression analysis. Lastly, the study used the Adjusted Hazard Ratio with a p-value < 0.05 and a 95% confidence interval to describe the connection.

### Results

The median duration for recovery was found to be 42 days with an interquartile range of 35 to 49. Children who received Amoxicillin had a four-times higher recovery rate (AHR = 4.09; 95% CI: 2.75, 6.07). The presence of diarrhea prolonged the recovery time by 53.0% (AHR = 0.47; 95% CI: 0.36, 0.62), while vomiting prolonged the recovery time by 58.0% (AHR =

**Data Availability Statement:** All relevant data is within the manuscript and its Supporting information files.

**Funding:** The author(s) received no specific funding for this work.

**Competing interests:** The authors have declared that no competing interests exist.

0.42; 95% CI: 0.32, 0.55). Edema reduced the chances of recovery by 48% (AHR = 0.52; 95% CI: 0.36, 0.62).

## Conclusion

The study found that recovery time for children with severe acute malnutrition is consistent with prior research. Key factors influencing recovery duration include the prompt administration of Amoxicillin upon admission and the presence of symptoms like diarrhea, vomiting, and edema. The findings emphasize the critical role of specific symptoms in predicting recovery times for children with SAM. By understanding these relationships, healthcare providers can enhance treatment strategies, improve resource management, and ultimately contribute to better health outcomes for affected children.

## 1. Introduction

Malnutrition is a multifaceted public health challenge that encompasses both undernutrition and overnutrition, often due to insufficient nutrient intake or poor absorption. Chronic malnutrition can lead to stunting and micronutrient deficiencies, sometimes referred to as "hidden hunger," which negatively impacts health, development, and economic stability, particularly during early childhood. Acute malnutrition, which arises from abrupt decreases in food intake or dietary quality, is frequently exacerbated by health issues [1, 2]. The World Health Organization (WHO) categorizes acute malnutrition into moderate acute malnutrition (MAM) and severe acute malnutrition (SAM). MAM is characterized by weight-for-height z-scores between -2 and -3 or a mid-upper arm circumference ranging from 115 to 124 mm, while SAM is defined by a weight-for-height z-score below -3, a mid-upper arm circumference of less than 115 mm, or the presence of bilateral pitting edema [2–5].

The 2023 UNICEF-WHO-WB Joint Child Malnutrition Estimates reveal troubling trends in child malnutrition from 2000 to 2022, with significant occurrences of stunting, wasting, overweight, and underweight among children under five. By 2022, approximately 148.1 million children globally were stunted, 45 million wasted, and 37 million overweight, with 45.5 million experiencing acute malnutrition each year, including 13 million severely affected [3]. The crisis is especially severe in Africa, with around 13.8 million children suffering from acute malnutrition [6, 7], contributing to approximately 1.7 million child deaths annually [7], This situation has been exacerbated by the COVID-19 pandemic, which added 6.7 million cases [6, 8]. SAM affects nearly 50 million children under five each year [6], leading to increased morbidity and mortality, as well as long-term developmental issues [7]. It disrupts essential physiological functions and requires careful management, particularly for those at risk of infections like diarrhea and pneumonia [1, 3]. SAM causes about one million child deaths annually, mainly in low- and middle-income countries [9], where severely malnourished children face a ninefold increase in mortality risk [1, 7]. In 2020, 13.6 million children experienced severe wasting, especially in Asia and sub-Saharan Africa [1, 3, 6, 9], with Ethiopia reporting some of the highest SAM rates [10]. A 2019 survey found that 37% of Ethiopian children aged 0–59 months were stunted [9, 11], 21% underweight [3, 7, 12, 13], and 7% wasted, with chronic malnutrition in Oromia reaching 38.2% [14].

Community-based management strategies are increasingly being used to combat SAM, with the WHO-supported Community-Based Management of Acute Malnutrition (CMAM)

program encouraging outpatient care through Outpatient Therapeutic Programs (OTP) [1, 15, 16] and utilizing Ready-to-Use Therapeutic Foods (RUTF), achieving nearly 80% recovery rates [1, 3, 7, 14–16]. CMAM targets children over six months without medical complications, focusing on improving coverage and treatment success while preventing future crises. Treatment outcomes are categorized as cured, defaulted, died, or non-respondent, which are crucial indicators of program success [7, 17]. Treatment outcomes are categorized as cured, defaulted, died, or non-respondent, which are crucial indicators of program success [7]. Despite being cost-effective [6], requiring about $2.6 billion for 90% coverage in high-burden regions, fewer than 20% of children with SAM currently receive adequate treatment [1, 6, 15, 18–21]. By 2020, coverage had risen to about 56%, with Outpatient Therapeutic Programs successfully treating 79–95% of uncomplicated SAM cases, achieving a recovery rate of 92% and a low mortality rate of 0.1% [7, 22]. However, progress toward the 2025 World Health Assembly nutrition targets and Sustainable Development Goal (SDG) 2.2 is insufficient, with only one-third of countries on track to halve childhood stunting by 2030 [7, 22–24]. Achieving the target of 89 million children with reduced stunting by 2030 will require more aggressive initiatives due to a projected shortfall of 39.5 million children, primarily in Africa [3]. Challenges such as research gaps, financial constraints, antibiotic resistance, and management of edematous acute malnutrition hinder treatment effectiveness [1–7, 14–21].

Ethiopia is enhancing its efforts to combat SAM through improved protocols, disease prevention, and economic development, alongside better food security monitoring [6, 25]. Although the prevalence of wasting decreased from 13% to 7% between 2016 and 2019, severe acute malnutrition rates have been rising due to factors such as droughts, civil unrest, and rising food prices, leading to an estimated economic loss of $230 million [7, 9]. A study in eastern Ethiopia found that children near OTP sites had significantly improved recovery times, particularly those with edema treated with amoxicillin, achieving a median recovery time of 49 days and a recovery rate of 79.8% [7]. Key recovery factors included the absence of comorbidities, timely access to treatment, referrals from trained mothers, and sufficient RUTF supplies [7, 9]. However, data on treatment success and obstacles is limited, especially regarding inpatient care before the COVID-19 pandemic, and localized data from emergency-prone regions is scarce [7, 26, 27]. This study aims to examine SAM by investigating recovery durations and identifying factors influencing outcomes, which are essential for guiding future healthcare policies and public health strategies in Ethiopia, particularly in drought-affected and food-insecure areas. It also seeks to provide valuable insights for researchers and the global scientific community in addressing this critical issue.

## 2. Materials and methods

### 2.1 Study area and period

The study was conducted in the Borena zone of the Oromia Region in Ethiopia. The zone consists of 13 woredas, with its capital city being Yabelo Town, located 567 km from Addis Ababa in the Southern part of the country. Borena is bordered by Kenya to the south, the SNNPR to the west, the West Guji and Guji zones to the north, and the Dawa zone of the Somali Region to the east. The zone has a total of five hospitals, 45 health centers, 165 health posts, and 70 private clinics. The population of the zone is 793,340, with 16.43% (130,346) of them being under five years old, and 15% (119,001) being children aged 6 to 59 months. From July 2021 to June 2022, a total of 6,005 children with severe acute malnutrition (SAM) were enrolled in an outpatient therapeutic feeding program in this zone [18]. Throughout the zone, there are 175 functional OTP sites that offer community-based treatment for severe acute malnutrition. The

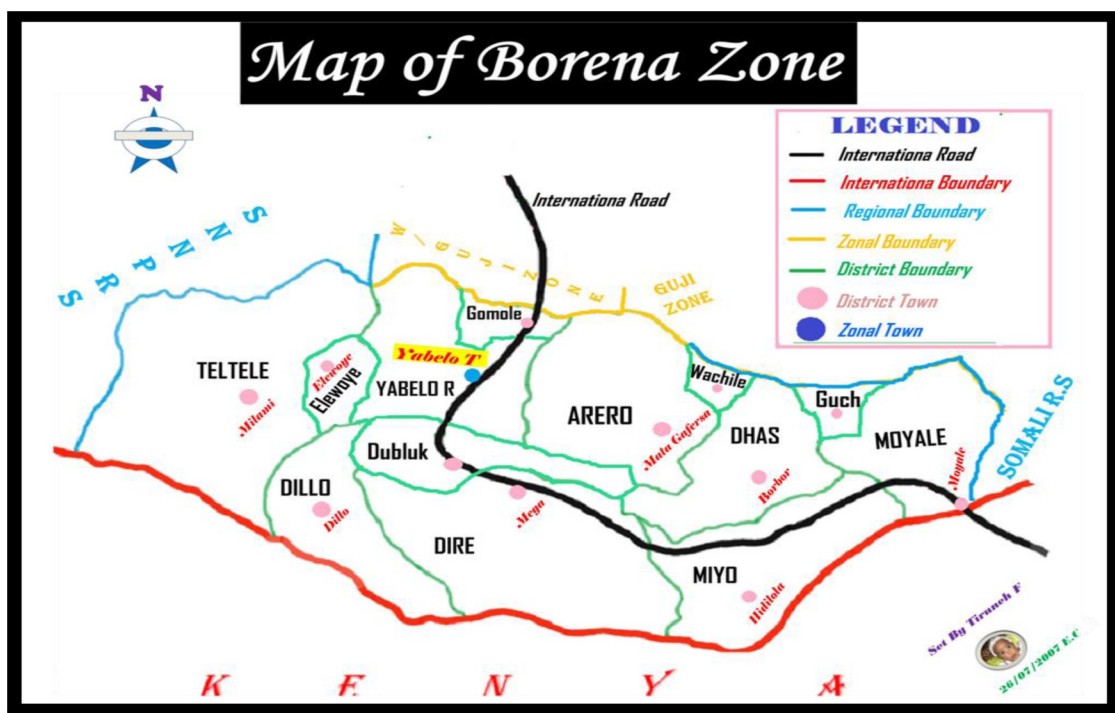

**Fig 1. Sketch map of Borena zone, Oromia region, Southern Ethiopia, 2023 G.C (Source: Borena zone administration office).**

study participants were enrolled from March 1–30, 2023, and followed weekly until June 30, 2023 (Fig 1).

## 2.2 Study design

A prospective follow-up design was employed in this institution-based study.

## 2.3 Study population and inclusion and exclusion criteria

The study population consisted of children aged 6 to 59 months in the Borena zone diagnosed with uncomplicated severe acute malnutrition, who passed an appetite test and were enrolled in the Outpatient Treatment Program (OTP). Participants were selected from randomly chosen OTP service sites during the study period, which spanned from March 1 to March 30, 2023. Eligible participants displayed bilateral pitting edema and had a Weight-for-Length/ Height (WFL/WFH) measurement of less than -3 Z-scores, with no medical complications and having passed the appetite test. Children admitted based on clinical discretion, those with congenital conditions impacting anthropometric measurements, those transferred from stabilization centers, and those with known chronic illnesses were excluded from the study. Notably, all children included in our research received only Ready-to-Use Therapeutic Food (RUTF) as part of their treatment within the OTP framework. Throughout the study period, no complementary treatments were provided, nor were any children receiving additional therapies included in our analysis. This focused methodology allowed for a clear assessment of the OTP's effectiveness in treating Severe Acute Malnutrition (SAM) without the confounding influences of other interventions.

**Table 1. Significant determinant of time to recovery from SAM among children 6–59 months used to calculate the largest sample size for the second objective.**

| Variable | % of outcome in unexposed group | AHR | Sample size by the second objective | Drop outs | Design effect | Final sample size | Reference |
|---|---|---|---|---|---|---|---|
| Mother travel < 30 minutes to service site | 19.9 | 2.6 | 82 | 5% | 1.5 | 129 | Amare B et al, 2021 [21]. |
| Children treated by trained Health Extension Workers (HEW) | 24 | 2.1 | 118 | 5% | 1.5 | 186 | |
| Taking Amoxicillin | 32.1 | 2.3 | 52 | 5% | 1.5 | 82 | Worku N et al, 2017. |
| Children with diarrhea | 88 | 0.81 | 204 | 5% | 1.5 | 322 | |

## 2.4 Sample size determination

To determine the sample size, we utilized the Epi Info version TM 7.2.5.0 software, employing a multi-stage sampling method and assuming a 95% confidence interval, 80% power, and a one-to-one exposed to unexposed ratio [20]. For the first objective, we used the program performance indicator of the frequency of children who recovered among those admitted (65.3%), as well as an adjusted hazard ratio (AHR) of 0.81 for the second objective, representing the effect of having diarrhea on the cure rate compared to not having diarrhea. Additionally, we considered the proportion of the outcome in the unexposed group, which was 88.18% among severely malnourished children enrolled in the OTP program in North Gondar, Ethiopia [20] Using the Epi Info software, we initially calculated a sample size of 204 for the second objective; however, we included a 5% compensation to account for potential loss to follow-up and adjusted for a design effect of 1.5 due to the multi-stage sampling method used. As a result, the final sample size for this study was determined to be 322. With this sample size, we anticipate having sufficient statistical power to identify significant associations between the predictors and the duration of recovery for children enrolled in the OTP program. (Table 1).

## 2.5 Sampling techniques and procedures

In the Borena zone, which comprises 13 districts and 175 functional OTP service sites, the sampling process was conducted in two stages. In the first stage, a lottery method was used to randomly select five districts from the total. Within these selected districts, there were 108 OTP sites. Moving to the second stage, 34 OTP sites were then randomly chosen from each of the selected districts using the same lottery method. This selection was carried out to ensure representativeness, targeting approximately 30% of the OTP sites within each district. Population proportion-to-size allocation was utilized to guarantee proportional representation. Due to the small scale of the study, all eligible children with severe acute malnutrition (SAM) were consecutively included from the randomly selected OTP sites. These children were followed up until the occurrence of the outcome, allowing for a comprehensive understanding of the duration of recovery for children in the OTP program and the factors influencing this period. By employing this sampling strategy, the study aims to encompass a diverse range of OTP sites and offer valuable insights into the population of children with severe acute malnutrition in the Borena zone (Fig 2).

## 2.6 Data collection, methods and instruments

The interviewer used a pre-tested, structured questionnaire and had a language expert translate it into Afan Oromo and back into English to ensure consistency. Data was collected by six trained diploma nurses with prior experience in data collection, who conducted in-person interviews with mothers or caregivers to gather information on socio-demographic characteristics

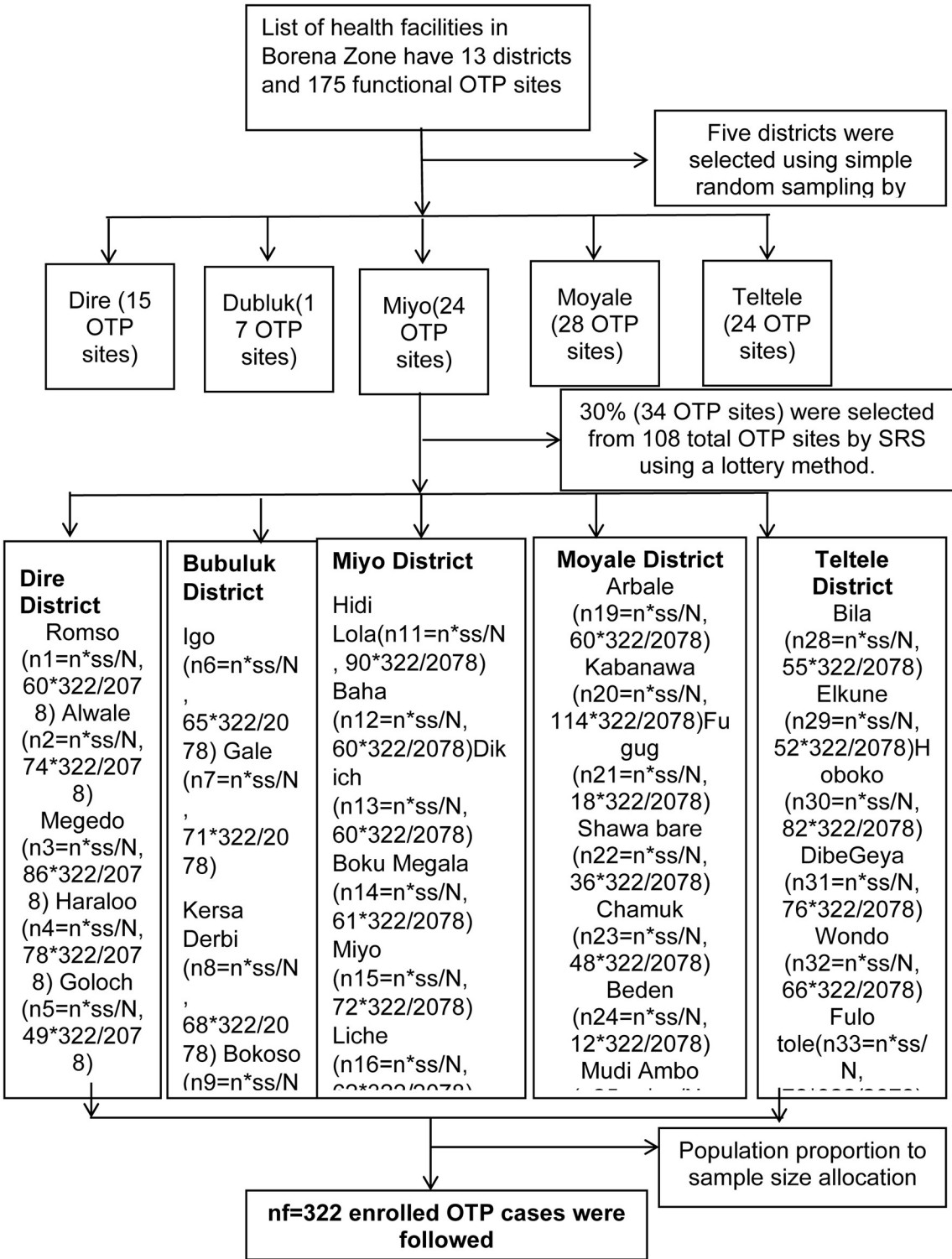

**Fig 2. Schematic diagram of sampling technique and procedure of duration of recovery and predictors, among 6 to 59 months age children registered at the OTP program in Borena zone, Oromia region, Ethiopia 2023.**

and household food variety. Anthropometric measures (weight and height/length) and physical examinations were used to identify signs of malnutrition in study participants. Weight was recorded weekly for children under two years using a calibrated 25 kg hanging scale, with measurements reported to the nearest 0.1 kg. A SECA scale (Société Européenne de Cardiologie et d'Athérosclérose) was used in our study to measure the anthropometric data, including weight and height, of children over two years old, ensuring regular calibration and accuracy checks to assess their nutritional status.Measurements of length and height were taken using a wooden measuring table and stadiometer, with specific procedures for different age groups. Medical complications and signs of malnutrition were assessed during enrollment and subsequent follow-ups using a standard OTP treatment card. The desire to eat assessment was conducted in a calm, private setting, and a child was considered to have passed the appetite test if they consumed part of the RUTF over thirty minutes [28]. Severely acute malnourished (SAM) children meeting specific criteria were included in the study and followed up weekly, with health and anthropometric data captured at each visit. Participants visited the site weekly for management and examination by the service provider and data collectors.

## 2.7 Study variables

**Dependent variable:** Time to recovery: measured in days from the date the SAM child admitted on OTP to the date the child was discharged as recovered (i.e. Absence of bilateral pitting edema for two uninterrupted visits AND WFH/WFL greater or equal to -2 z-score and clinically in good condition and attentive) is considered as dependent variables. Children who were discharged from treatment but had not fully recovered, such as those who were transferred out, defaulted, died, or did not respond, were considered as censored.

**Independent variables:**

**Socio- demographic factors** include age, sex, residence place, mother's education level, mother's occupation, marital status, distance from OTP site, house hold food security status, and family size;

**Co-morbid conditions (presence):** Diarrhea, vomiting, cough, skin infection, blood in stool, anemia (palmar pallor);

**Routine medication:** Providing Amoxicillin, de-worming tablet, folic acid; the treatment protocols followed in our study adhered to the standardized guidelines established by the World Health Organization for the management of severe acute malnutrition in children. The dosage of Amoxicillin, specifically, was determined based on the weight of the children and followed the recommended guidelines for treatment of infections common in this population. The doses were carefully calculated for each child based on their weight and clinical condition at the time of treatment. For instance, children typically received 20–40 mg/kg/day of Amoxicillin divided into two doses. This approach ensures that each child receives an appropriate dosage tailored to their specific needs. We maintained detailed records of the medication administered, including dosages, to ensure compliance with treatment protocols and the ability to assess any impact on recovery times.

**Types of severe acute malnutrition:** Edematous, Non-edematous (Marasmus);

**Service provision related variables:** Key message provision, SAM child referral way to OTP

## 2.8 Operational definition

**OTP admission criteria:** Bilateral pitting edema (grade + or ++) OR WFL/WFH less than -3 Z-score AND absence of medical complications, passed desire to eat and clinically good and vigilant [25].

**Routine medication:** Children older than two years will receive deworming tablet during their second visit. If anemia symptoms are present, a 5 mg dose of folic acid will be given once and amoxicillin will be administered upon admission for all SAM children for seven days [25, 29].

**Recovered**: is cured fully in OTP- (If admitted by edema: Absence of bilateral pitting edema for 2 successive visits and WFH/WFL greater or equal to -2 z-score and clinically good and vigilant. If enrolled based on WFH/WFL, discharged cured when: WFH/WFL greater or equal to -2 z-score, absence of bilateral pitting edema and clinically good and vigilant) [25, 29].

**Event**: Refers to the recovery of severely malnourished children while discharged from the program.

**Censored:** in this study included malnourished children who did not recover due to transfer out, default, death, or non-responding.

**Time to recovery**: was measured in days from the child admitted by SAM on OTP to the child discharged as recovered [25, 29].

**Chronic diseases**: TB, HIV, DM, Chronic heart diseases [28]. The Household Food Diversity Score (HHFD):The HHFD specifically measures dietary diversity by assessing the variety of food groups consumed within a household. This score emphasizes the breadth of food intake rather than the quantity consumed, thereby providing valuable insights into nutritional quality. Typically, the HHFD evaluates the consumption of specific food groups, such as grains, vegetables, and proteins, to determine dietary diversity levels. In our study, the HHFD was measured using a standardized tool that examines the intake of seven distinct food groups. Households reported their dietary consumption over a specified period, and scores were computed using SPSS software. Based on the scores, households were classified into categories: those achieving a score of 4.5 or higher were classified as having medium dietary diversity, while those scoring below 4.5 were categorized as having poor dietary diversity. This threshold was established based on a combination of established literature and the specific characteristics of our study population. The selection of 4.5 as a cutoff aligns with thresholds utilized in previous research that has validated similar measures for food security. Numerous studies have shown that a score of 4.5 effectively differentiates between food secure and food insecure households across diverse populations [19, 25, 28–35]. This threshold is widely referenced in food security assessments and has been validated in various contexts. Additionally, our statistical analysis indicated that the median HHFS score among participants was approximately 4.5. By adopting this value, we ensure that our classification accurately reflects the distribution of food security statuses within our sample, thereby enhancing the relevance and applicability of our findings to the children in the Borena Zone. Utilizing this specific cutoff point provides a clear framework for policy implications and interventions aimed at addressing food insecurity among vulnerable populations, reflecting a pragmatic approach to understanding household food access and its relationship with severe acute malnutrition [19, 25, 28–35].

**Key Messages of OTP:**—Deliver the following instructions to the caregivers: inform them about the appropriate timing and method of administering medication to the child, guide them on how to give Ready-to-Use Therapeutic Food (RUTF) for the child, educate them on day to return to outpatient care, and emphasize the importance of taking the child to the hospital immediately if their condition worsens [25].

## 2.9 Data processing and analysis

The data collected was coded, edited, and entered into Epi data version 3.1 before being exported and analyzed using SPSS version 26. Descriptive analysis was carried out to assess missing data, outliers, normality, and multi-collinearity. The Kaplan-Meier survival curve was

used to compare recovery times among different independent categorical variables, and the model's fitness was evaluated using the log-rank (Mantel-Cox) test and graphical assessment of proportional hazards over time. Cox regression analysis was utilized to identify factors affecting recovery time from severe acute malnutrition, with adjusted Hazard Ratio and 95% confidence interval determining significant associations. The study's findings were presented through tables, figures, charts, and descriptions.

### 2.10 Data quality assurance

To ensure data quality, the collection tool was adapted to address study variables, and a pre-test was conducted on 5% of the sample in another district before actual data collection. Training was provided to data collectors and supervisors, and data collection was closely monitored by the principal investigator and supervisors. Thorough review and daily checks were conducted to ensure data completeness and accuracy.

### 2.11 Ethical consideration

Prior to initiating our study, we secured ethical approval from the Dila University's College of Health Sciences and Medicine Institutional Review Board (IRB), which approved our approach to obtaining verbal consent, considering the nature of the research and participant demographics. The approval number is "[DUCHM/IRB/011/2023]," granted on "January 22, 2023." Following this, the IRB coordinator communicated the study's approval to the Borena Zone Health Office, which relayed the information to the relevant District Health Offices and Outpatient Therapeutic Program (OTP) sites. To document the verbal consent process accurately, we provided participants—primarily mothers or caregivers of the children—with a detailed oral explanation of the study's purpose, procedures, potential risks, and benefits. Participants were encouraged to engage in a dialogue to ensure comprehension, while a trained research assistant witnessed and recorded their consent, capturing confirmation and understanding alongside the participant's identification number and date of consent. A secure log was maintained, documenting participants who provided verbal consent, which included identification numbers and a summary of the consent discussions, ensuring adherence to data management protocols for confidentiality and protection. We underscored the importance of confidentiality throughout the consent process and detailed the security measures in place for safeguarding participants' data. Notably, our data collection forms did not contain any personally identifiable information regarding the children enrolled in the OTP, thereby preserving their privacy. To further ensure anonymity, we employed unique identification numbers assigned during the OTP registration to correlate baseline information with outcomes, while ensuring no personally identifiable information was disclosed. We clearly articulated the study's objectives to optimize participant understanding and create an environment that encouraged honest responses, ultimately aiming to mitigate the risk of social desirability bias.

## 3. Results

### 3.1 Socio-demographic characteristics of participants

The study involved 322 individuals who all participated, resulting in a 100% response rate. The average age of the study subjects was 22.6 months, with a standard deviation of ± 14.6. Out of the 322 participants, 211 (65.5%) were aged < = 24 months, and slightly over half, 167 (51.9%), were male. The majority of the participants, 265 (82.3%), resided in rural areas. In terms of family size, most children, 290 (90.1%), belonged to families with less than 5

**Table 2. Socio-demographic characteristics of caretakers and SAM managed at OTP in Borena zone, Southern Ethiopia 2023 (n = 322).**

| Variables | Frequency | % |
|---|---|---|
| **Sex** | | |
| Male | 167 | 51.9 |
| Female | 155 | 48.1 |
| **Age in month** | | |
| ≤ 24 | 211 | 65.5 |
| > 24 | 111 | 34.5 |
| **Place of residence** | | |
| Rural | 265 | 82.3 |
| Urban | 57 | 17.7 |
| Family size | | |
| < 5 | 290 | 90.1 |
| ≥ 5 | 32 | 9.9 |
| **Caretakers educational status** | | |
| Unable to read and write | 155 | 48.1 |
| Primary | 51 | 15.8 |
| Secondary and above | 116 | 36 |
| **Marital status of the mother/caregiver** | | |
| Married | 294 | 91.3 |
| Single | 9 | 2.8 |
| Divorced | 9 | 2.8 |
| Widowed | 10 | 3.1 |
| **Mother's/caregivers' occupation** | | |
| Housewife | 220 | 68.3 |
| Farmer | 32 | 9.9 |
| Government employee | 46 | 14.3 |
| Merchant | 24 | 7.5 |
| **Travelling distance to the health post** | | |
| <30 minutes | 269 | 83.5 |
| > = 30 minutes | 53 | 16.5 |

members. Additionally, more than 82.3% of the participants traveled less than 30 minutes to reach OTP sites (Table 2).

## 3.2 Household Food Diversity status of Caretakers/mothers and children

The analysis of household food diversity revealed that a significant number of households included grains/cereals and white tubers and roots in their diet. Nevertheless, there was a noticeable lack of consumption when it came to pulses, nuts, seeds, milk and milk products, as well as other vegetables, as indicated in (Table 3).

## 3.3 Routine medications provision

In this context, 279 children participating in the research were administered amoxicillin upon admission, representing 86.6% of the total. Conversely, only 28 children, amounting to 14.3%, were given deworming medication during the subsequent monitoring period. These results can be found in (Table 4).

**Table 3. Household food diversity score of caretakers and children aged 6–59 months diagnosed with SAM managed at OTP ram in Borena zone, Southern Ethiopia 2023 (n = 322).**

| Variables | Frequency | % |
|---|---|---|
| Food made from grains/cereals and White tubers and roots | | |
| Yes | 284 | 88.2 |
| No | 38 | 11.8 |
| Pulses, Nuts and seeds (tree nuts and ground) | | |
| Yes | 59 | 18.3 |
| No | 263 | 81.7 |
| Milk and milk products | | |
| Yes | 187 | 58.1 |
| No | 135 | 41.9 |
| Organ meat (iron-rich), Flesh meats, Eggs and food made from it. | | |
| Yes | 148 | 46 |
| No | 174 | 54 |
| Dark green leafy vegetables | | |
| Yes | 71 | 22 |
| No | 251 | 78 |
| Vitamin-A rich vegetables and fruits | | |
| Yes | 91 | 28.3 |
| No | 231 | 71.7 |
| Other vegetables* | | |
| Yes | 102 | 31.7 |
| No | 220 | 68.3 |
| The overall HHFD score | | |
| <4.5 (Poor) | 243 | 75.5 |
| ≥ 4.5 (Medium and above) | 79 | 24.5 |

*****O**ther vegetables: refers to a variety of non-starchy vegetables, including but not limited to leafy greens, tomatoes, carrots, cauliflower, and other similar produce. This category encompasses vegetables that are not classified under common types, such as root vegetables or legumes, to provide a more comprehensive understanding of the dietary diversity assessed in our study.

**3.3.1 Co-morbidity conditions.** Moving on to the co-morbidity conditions, it was observed that 63.7% of the children had acute diarrhea during their admission, while 49.4% experienced vomiting. Furthermore, 31.4% of the children displayed signs of edema upon admission. These findings are presented in (Table 4).

**3.3.2 Types of severe acute malnutrition during admission (n = 322).** Severe acute malnutrition cases were classified into two types during admission, with 221 cases (68.6%) diagnosed with marasmus and the remaining 101 cases (31.4%) diagnosed with edema, as illustrated in (Fig 3).

## 3.4 Survival status of children

The median follow-up period for children undergoing OTP treatment for severe acute malnutrition was 49 days, ranging from 21 to 112 days. The median recovery duration was 42 days, with an interquartile range of 35 to 49 days. The survival probability during OTP treatment was high, starting at approximately 0.99 on the third day, decreasing to 0.01 by the end of the 112-day follow-up period, as shown in (Fig 4).

**Table 4. Co-morbidities and routine medications provided for children aged 6–59 months diagnosed with SAM managed at OTP ram in Borena zone, Southern Ethiopia 2023 (n = 322).**

| Variables | Frequency | % |
|---|---|---|
| Amoxicillin given during admission | | |
| Yes | 279 | 86.6 |
| No | 43 | 13.4 |
| Deworming | | |
| Yes | 28 | 14.3 |
| No | 88 | 85.7 |
| Diarrhea during admission | | |
| Yes | 205 | 63.7 |
| No | 117 | 36.3 |
| Vomiting during admission | | |
| Yes | 159 | 49.4 |
| No | 163 | 50.6 |
| Cough during admission | | |
| Yes | 129 | 40.1 |
| No | 193 | 59.9 |
| Blood in stool | | |
| Yes | 2 | 0.6 |
| No | 320 | 99.4 |
| Palmar pallor | | |
| Yes | 16 | 5.0 |
| No | 306 | 95.0 |
| Skin infections | | |
| Yes | 235 | 73.0 |
| No | 87 | 27.0 |

Analysis through the log-rank test demonstrated that children with skin infections had a significantly longer recovery time during OTP treatment compared to those without skin infections (p < 0.01), as depicted in (Fig 5).

Furthermore, the log-rank test revealed that children who received Amoxicillin upon admission for OTP treatment had a significantly shorter recovery time than those who did not receive it (p < 0.01), as shown in (Fig 6).

## 3.5 Treatment outcomes of children with severe acute malnutrition

In the study on treatment outcomes of children with severe acute malnutrition, it was found that 93.2% of the total study subjects successfully recovered from the condition. There were 1.9% cases of default, 2.5% cases of non-response, 0.3% cases of mortality, and 2.2% cases of transfer out.

## 3.6 Predictors of time to recovery

The bivariate analysis identified several factors that can predict the time to recovery. These factors include travel distance from the OTP site, delivery of OTP key messages, household food diversity score, administration of Amoxicillin during admission, deworming, presence of diarrhea, vomiting, skin infection, and edema at admission. A statistical Log rank (Mantel-Cox) test was conducted to check the model appropriateness, which showed a significant difference

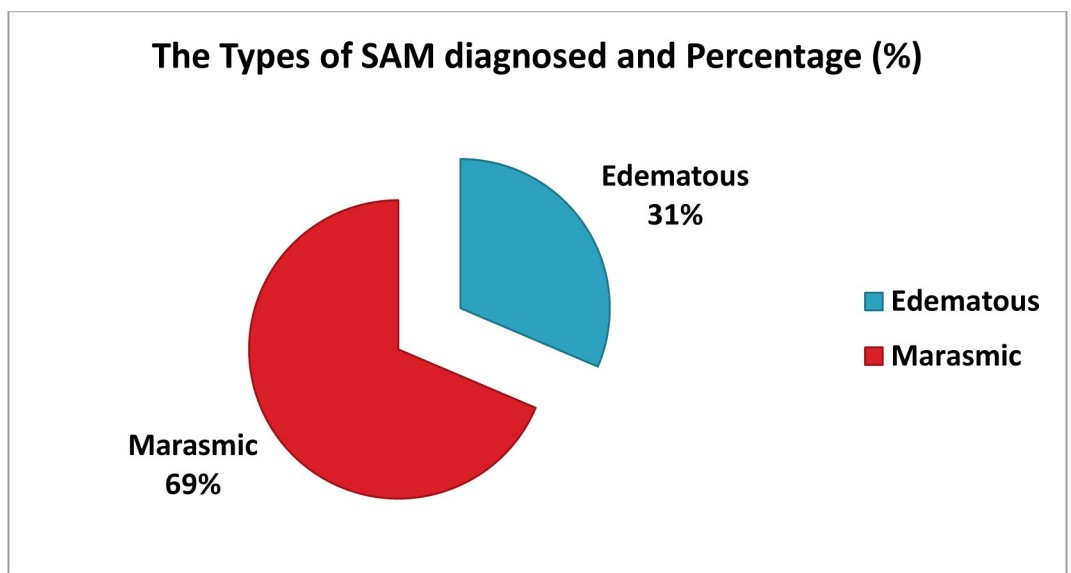

**Fig 3. Types of severe acute malnutrition diagnosed during admission among children aged 6–59 months with SAM managed at OTP in Borena zone, Southern Ethiopia 2023.**

in survival between the groups with a significance level of P<0.001. After adjusting for possible confounders in the multivariable analysis, it was found that children who received Amoxicillin during admission had a 4.09 times higher rate of recovery (AHR = 4.09; 95% CI: 2.75, 6.07) compared to those who did not receive it. Children with diarrhea at admission had a 53% longer recovery time (hazard ratio = 0.47; 95% CI = 0.36, 0.62) compared to those without diarrhea. On the other hand, children with vomiting at admission had a 58.0% longer recovery rate (AHR = 0.42; 95% CI: 0.32, 0.55) compared to their counterparts. The presence of edema at admission reduced the cure rate by 48% (AHR = 0.52; 95% CI: 0.36, 0.62) (Table 5).

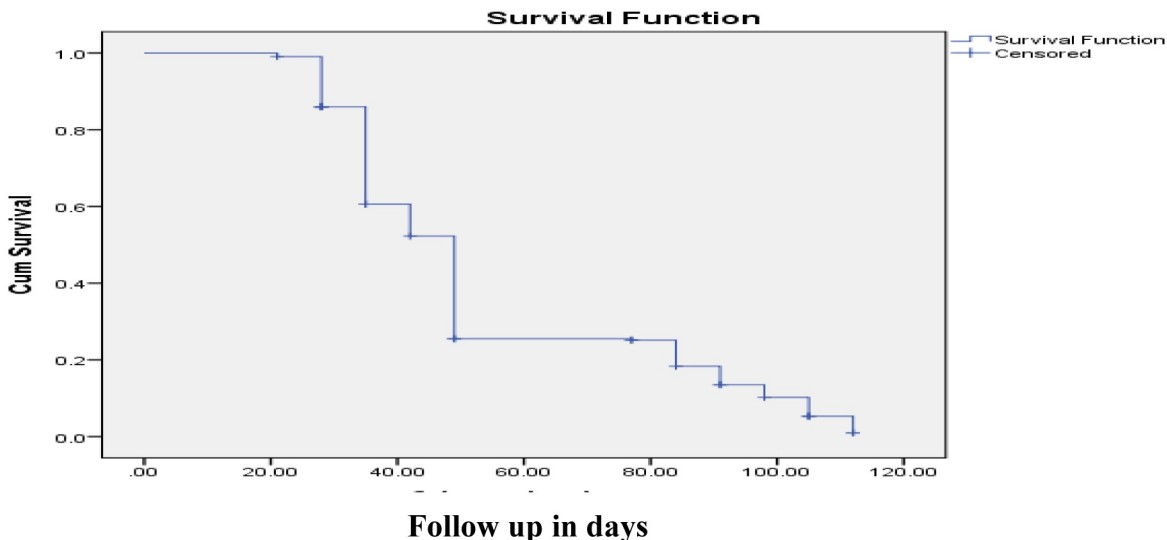

**Fig 4. Survival probability on OTP treatment among children 6 to 59 months of age diagnosed with SAM managed at OTP in Borena zone, Oromia, Southern Ethiopia 2023.**

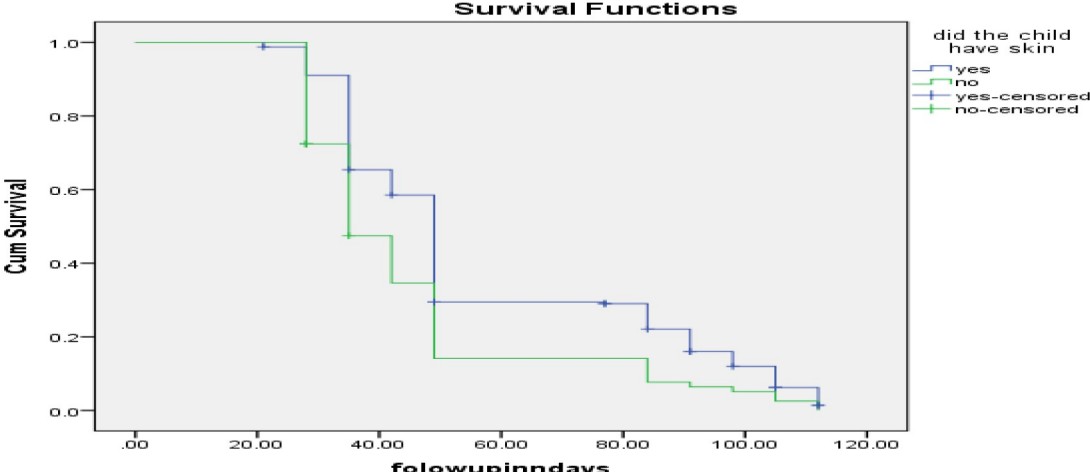

**Fig 5. Survival status among SAM children 6 to 59 months of age with skin infection with managed at OTP in Borena zone, Oromia, Southern Ethiopia 2023.**

## 4. Discussion

This study aimed to determine the median recovery time from severe acute malnutrition (SAM) and evaluate factors affecting recovery in the Borena zone of Oromia region, Southern Ethiopia, in 2023. Of the 322 monitored children, 93.2% recovered from SAM, indicating the effectiveness of the treatment protocols. The median recovery time was 42 days, with an inter-quartile range (IQR) of 35 to 49 days, and the longest recovery time recorded was 112 days, with no children exceeding this duration. Key factors influencing recovery included the presence of symptoms such as diarrhea, vomiting, and edema at admission, which were linked to longer recovery times. Conversely, the timely administration of routine medications like Amoxicillin was associated with quicker recovery rates.

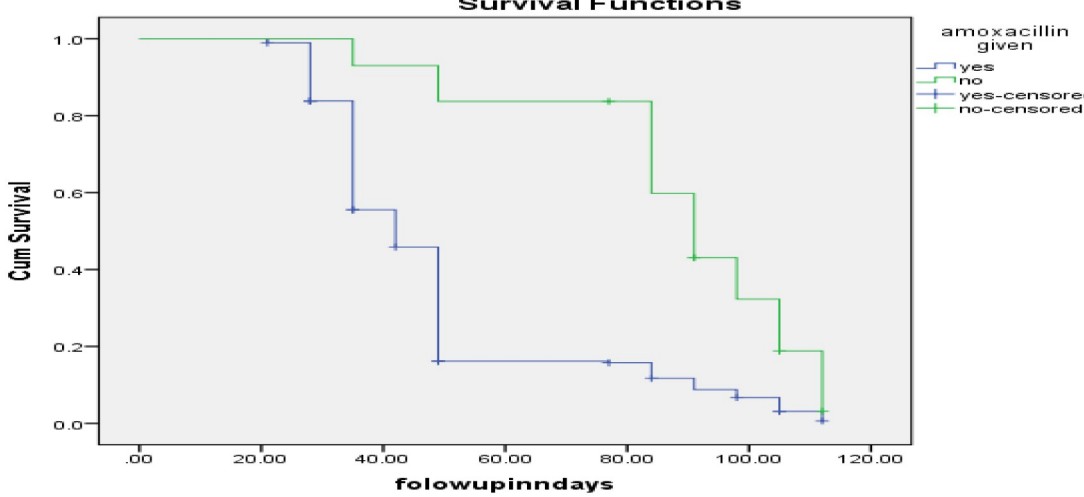

**Fig 6. Survival status among SAM children who received Amoxicillin while OTP management in Borena zone, Oromia, Southern Ethiopia 2023.**

**Table 5. Predictors of time to recovery among children aged 6–59 months diagnosed with SAM managed at OTP in Borena zone, Oromia, Southern Ethiopia 2023 (n = 322).**

| Variables | Outcome | | CHR(95%:CI) | AHR(95%:CI) | P-value |
|---|---|---|---|---|---|
| | Event | Censored | | | |
| **Traveling distance to Health post** | | | | | |
| < 30 minute | 251 | 18 | 1.45(1.07, 1.97) | 0.96(0.69, 1.34) | 0.83 |
| >/ = 30 minute | 49 | 4 | 1 | 1 | |
| **Key messages delivered** | | | | | |
| Yes | 219 | 16 | 0.65(0.50,0.84) | 1.61(0.22,11.82) | 0.64 |
| No | 81 | 6 | 1 | 1 | |
| **HHFDS** | | | | | |
| <4.5 | 226 | 17 | 0.76(0.58,0.98) | 0.97(0.74,1.28) | 0.85 |
| >/ = 4.5 | 74 | 5 | 1 | 1 | |
| **Amoxicillin at admission** | | | | | |
| Yes | 262 | 17 | 2.48(1.75,3.53) | 4.09(2.75, 6.07) | < 0.001 |
| No | 38 | 5 | 1 | 1 | |
| **Deworming during follow up** | | | | | |
| Yes | 20 | 8 | 1.23(0.87, 1.70) | 0.87(0.61,1.23) | 0.42 |
| No | 76 | 12 | 1 | 1 | |
| **Diarrhea at admission** | | | | | |
| Yes | 190 | 15 | 0.48(0.38,0.62) | 0.47(0.36,0.62) | <0.001 |
| No | 110 | 7 | 1 | 1 | |
| **Vomiting at admission** | | | | | |
| Yes | 149 | 10 | 0.75(0.59,0.94) | 0.42(0.32, 0.55) | <0.001 |
| No | 151 | 12 | 1 | | |
| **Skin infections at admission** | | | | | |
| Yes | 219 | 16 | 0.64(0.49,0.83) | 0.85(0.64, 1.13) | 0.27 |
| No | 81 | 6 | 1 | | |
| **Edema present at admission** | | | | | |
| Yes | 55 | 7 | 0.57(0.42,0.77) | 0.52(0.36,0.62) | <0.001 |
| No | 245 | 15 | 1 | | |

The study results indicate that the median recovery time for children undergoing treatment for severe acute malnutrition (SAM) was 42 days, with an interquartile range (IQR) of 35 to 49 days. Notably, the longest recovery time observed in this study was 112 days, with no child exceeding this duration. This reflects the success of the intervention strategies and signifies effective SAM management within therapeutic feeding units. Notably, the median recovery time met and exceeded the international standard of a maximum average length of stay of 60 days for therapeutic feeding units, underscoring the treatment protocols' efficacy. These findings align with the National Guidelines for SAM management in Ethiopia, which emphasize timely interventions, and are consistent with global reference standards established by the Sphere Project [28]. Additionally, the study supports findings from various regional studies in Ethiopia [7, 20, 36–39] and in Africa [22] that document comparable recovery times and outcomes. Research in different Ethiopian regions has shown similar recovery metrics, reinforcing the notion of effective intervention strategies. For instance, Yadeta et al. (2024) reported a cure rate of 89.6% with a median length of stay of 7 weeks in eastern Ethiopia [7]. In Southwest Ethiopia, Wondie et al. (2022) found a recovery rate of 54.4% with a median recovery time of 49 days [36]. Institution-based prospective cohort study conducted at public health institutions

in Afar Regional State, Ethiopia by Liben et al. 2019 revealed that the mean length of stay was 44.15(± 8.77) days [39]. Moreover, a study in Accra, Ghana, by Takyi et al. (2021) found that among children who achieved cure, the mean time to cure was 9.4 weeks (median 7.5, range 1–26), with a cure rate of only 34.5% [22]. Other studies revealed median recovery times of 8.7 weeks [37], 49 days [38], 38.5 days [20],and 44.15(± 8.77) days [39], consistently demonstrating successful outcomes across different settings. Moreover, a study in Accra, Ghana, by Takyi et al. (2021) found that among children who achieved cure, the mean time to cure was 9.4 weeks (median 7.5, range 1–26), with a cure rate of only 34.5% [22]. The comparability of these findings is attributed to similar methodologies for assessing recovery, adherence to international guidelines, and effective intervention strategies. Furthermore, the interconnectedness of health systems within Ethiopia facilitates the sharing of knowledge and resources, enhancing the effectiveness of interventions.

In contrast, a recent study by Feleke et al. (2024) in Northeast Ethiopia reported a longer median recovery time of 56 days and a recovery rate of 74.7% [29]. Kitesa et al. (2023) found a median recovery time of 49 days in the Borena Zone, indicating longer durations compared to our findings [9]. Conversely, an institutional-based multicenter retrospective follow-up study by Eyi et al. (2022) in Southwest Ethiopia indicated that recovery from severe acute malnutrition (SAM) was below the accepted minimum standard, even after a maximum treatment duration of 60 days [40]. A study by Baraki et al. (2020) in the Amhara region documented a notably shorter median recovery time of 16 days [41] as well as Teshome et al. (2019) recorded a median recovery time of 36 days in Shebedino, Southern Ethiopia which is considerably lower than our study's findings [8] Variability in reported recovery times can be attributed to differences in study design, sampling techniques, criteria for defining recovery, and demographic characteristics of the populations studied. Factors such as baseline nutritional status, comorbidities, cultural attitudes toward nutrition, and environmental conditions all influence recovery rates.

The findings carry significant implications for clinical practice, public health policy, and research. The variability in recovery durations necessitates tailored interventions and resource allocation based on local needs, especially in areas with prolonged recovery times. Additionally, inconsistencies in recovery definitions across studies highlight the need for standardizing metrics to improve data comparability and reliability. Recognizing the influence of comorbidities emphasizes the importance of comprehensive care that addresses both nutritional and health conditions. Culturally sensitive approaches and community involvement are crucial for enhancing treatment adherence and program effectiveness. Continuous evaluation of treatment protocols is essential for refining nutritional programs and adopting best practices. Finally, addressing disparities in healthcare access is vital for strengthening infrastructure in areas with high SAM prevalence, necessitating further research to investigate the factors affecting variations in recovery times.

The study's results indicate that certain symptoms, particularly diarrhea at admission, significantly predict recovery times for children with severe acute malnutrition (SAM). This aligns with various research in Ethiopia and Ghana that underscores the importance of these clinical indicators. For example, Yadeta et al. (2024) found that the absence of diarrhea at admission is an independent predictor of recovery (AHR of 1.51; 95% CI: 1.18–1.94) in eastern Ethiopia [7]. Similarly, Wondie et al. (2022) [36] reported that diarrhea at admission significantly influences recovery times in a retrospective cohort study conducted in Southwest Ethiopia. Gebremedhin et al. (2020) [38] also identified diarrhea at admission as an independent predictor of recovery time in Arba Minch Zuria Woreda, Southern Ethiopia. Furthermore, Takyi et al. (2021) [22] recognized diarrhea as a predictor of recovery time in their study in Accra, Ghana. Additional studies by Teshome et al. (2019) [8] and Mamo et al. (2019) [20] in

Southern and Northwest Ethiopia, respectively, corroborate these findings, highlighting the effects of diarrhea and other comorbidities on recovery duration. The consistent results across studies suggest that diarrhea is a critical clinical indicator affecting recovery times globally, linked to its physiological impact on fluid and nutrient absorption, as well as common healthcare challenges in Ethiopia and other African nations. Many studies employed similar methodologies, enhancing the reliability of their findings. Diarrhea often correlates with other health issues like infections and poor nutritional status, which are prevalent among children with SAM. Monitoring diarrhea at admission is crucial, enabling healthcare providers to identify children at higher risk for prolonged recovery and implement timely interventions. Recognizing diarrhea as a key recovery predictor can inform treatment protocols and guide healthcare training programs to address symptoms and comorbidities in children with SAM. On a broader scale, results can inform public health initiatives aimed at reducing SAM incidence and its complications, emphasizing the importance of improving sanitation, access to clean water, and nutritional education in vulnerable populations.

In line, the study found a significant association between edema at admission and recovery time in children with SAM, reflecting findings from other Ethiopian studies [7, 37, 38]. Yadeta et al. (2024) reported that edema at admission (AHR = 1.74; 95% CI: 1.33–2.29) serves as an independent predictor of recovery [7]. Similarly, Baraki et al. (2020) [42] and Gebremedhin et al. (2020) [38] confirmed the significance of edema in recovery time from SAM in Northwest and Southern Ethiopia, respectively. Additionally, Mamo et al. (2019) indicated that children exhibiting vomiting at admission significantly predict recovery time [20]. The consistency of these findings suggests that edema and vomiting are recognized clinical indicators of malnutrition severity. Factorial similarities may arise from consistent methodologies, definitions, and diagnostic criteria, reinforcing comparability of results across regions facing similar public health challenges related to infections and malnutrition. The findings highlight the importance of closely monitoring these symptoms at admission for timely interventions, and recognizing edema and vomiting can guide targeted treatment protocols, improving recovery outcomes for children with SAM. Training healthcare providers to identify and manage these symptoms can enhance clinical decision-making, while understanding recovery predictors informs resource allocation and prioritizes interventions.

Furthermore, the study emphasizes the significance of antibiotic treatment, particularly amoxicillin, as a recovery predictor in SAM. Yadeta et al. (2024) found amoxicillin administration (AHR = 1.55; 95% CI: 1.19–2.02) critical for recovery in eastern Ethiopia [7], and similar results were noted by Mamo et al. (2019) [20] and Fikrie et al. (2019) [43], confirming the effectiveness of antibiotics in improving recovery times. The common context of significant public health challenges likely leads to similar findings regarding antibiotic treatments, particularly their positive impact on recovery outcomes. Methodological consistency enhances the reliability of results, emphasizing the need for integrating antibiotic treatment into clinical guidelines for SAM management [7, 20, 39, 43, 44]. Recognizing and treating infections in malnourished children, along with ensuring the availability of critical medications, is crucial for improving recovery rates. The findings advocate for a dual approach focusing on nutritional support and infection management, promoting effective treatment strategies. Further research is warranted to explore the mechanisms by which antibiotics influence recovery, including optimal timing and dosage to maximize effectiveness.

Finally, this study uses primary data to analyze causal relationships but faces challenges in participant follow-up, leading to missing data and reduced statistical power. To mitigate these issues, various strategies were implemented, including a structured follow-up schedule with reminders and active caregiver involvement to enhance retention rates. The importance of caregivers in tracking their child's progress was reinforced through small incentives for

attendance and flexible appointment scheduling to meet their needs. To address missing data, appropriate statistical techniques, such as imputation methods, were employed to maintain analysis integrity. Comparative analyses were also conducted to identify potential biases between participants who completed follow-ups and those who did not. While efforts were made to reduce social desirability bias that could affect the accuracy of dietary variety data from the 24-hour food recall, this issue persists. Measures were specifically taken during data collection to minimize this bias. Researchers communicated the study's objectives clearly to mothers and caregivers, emphasizing the assessment of treatment effectiveness and the improvement of health outcomes for children with severe acute malnutrition. This transparent communication aimed to create a collaborative environment that encouraged honest responses, with participants assured of confidentiality and that their information would be used exclusively for research purposes, thus alleviating concerns about potential judgment or negative consequences.

## 5. Conclusion

The study demonstrates that treatment protocols for severe acute malnutrition (SAM) in the Borena zone of Oromia region, Ethiopia, are effective, with a high recovery rate of 93.2% and a median recovery time of 42 days that aligns well with international standards. It identifies critical clinical factors—diarrhea, vomiting, and edema at admission—as significant predictors of recovery time, underscoring the importance of monitoring these symptoms for improved management of SAM. The findings highlight the necessity for localized interventions tailored to specific recovery challenges and resource availability in various regions. They also reveal inconsistencies in recovery definitions across studies, advocating for standardized metrics to enhance data comparability and reliability of outcomes. The study emphasizes the influence of comorbidities on recovery, recommending holistic care strategies that integrate medical treatment, nutritional support, and community involvement. Furthermore, the study suggests enhancing healthcare provider training to better identify and manage key symptoms, ultimately improving clinical decision-making and patient outcomes. It supports public health initiatives aimed at improving sanitation, access to clean water, and nutritional education in areas with a high prevalence of SAM. The findings also call for further research to explore the mechanisms by which antibiotics and other treatments affect recovery, alongside a comprehensive understanding of socio-economic and cultural factors influencing malnutrition outcomes. Effective management of SAM will require prudent resource allocation to ensure sufficient funding and support for interventions, particularly in high-prevalence zones with limited healthcare access.

## Supporting information

**S1 Table. English version questionnaires.**
(DOCX)

## Acknowledgments

We would like to thank Dilla University College of Health and Medical Science, School of Public health, Reproductive Health Department for giving us this opportunity and for the overall support provided to undertake this thesis. We would also like to thank Borena Zone Health Office for supplying important data. Finally, but just as particularly, we thank the data collectors, supervisors and study participants for their remarkable input for this study.

## Author Contributions

**Conceptualization:** Girma Tenkolu Bune, Abuna Mohammed.

**Data curation:** Girma Tenkolu Bune, Abuna Mohammed, Samrawit Hailu, Eden Ashenafi.

**Formal analysis:** Girma Tenkolu Bune, Abuna Mohammed.

**Funding acquisition:** Abuna Mohammed, Samrawit Hailu.

**Investigation:** Girma Tenkolu Bune, Abuna Mohammed.

**Methodology:** Girma Tenkolu Bune, Abuna Mohammed, Eden Ashenafi.

**Project administration:** Abuna Mohammed, Samrawit Hailu, Eden Ashenafi.

**Resources:** Abuna Mohammed, Samrawit Hailu, Eden Ashenafi.

**Software:** Girma Tenkolu Bune, Abuna Mohammed, Samrawit Hailu, Eden Ashenafi.

**Supervision:** Girma Tenkolu Bune, Samrawit Hailu, Eden Ashenafi.

**Visualization:** Abuna Mohammed, Samrawit Hailu.

**Writing – original draft:** Abuna Mohammed.

**Writing – review & editing:** Girma Tenkolu Bune, Samrawit Hailu, Eden Ashenafi.

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
