## [Decision Letter · Decision Letter 0]

16 May 2024

PONE-D-24-07649Recovery Time and Predictors of Severe Acute Malnutrition in Children Aged 6–59 Months via an Outpatient Therapeutic Program in Borena Zone: A Prospective Study.PLOS ONE

Dear Dr. Bune,

Thank you for submitting your manuscript to PLOS ONE. After careful consideration, we feel that it has merit but does not fully meet PLOS ONE’s publication criteria as it currently stands. Therefore, we invite you to submit a revised version of the manuscript that addresses the points raised during the review process.

We look forward to receiving your revised manuscript.

Kind regards,

Ammal Mokhtar Metwally, Ph.D (MD)

Academic Editor

PLOS ONE

Journal Requirements:

2. In the ethics statement in the Methods, you have specified that verbal consent was obtained. Please provide additional details regarding how this consent was documented and witnessed, and state whether this was approved by the IRB.

3. We note that your Data Availability Statement is currently as follows: "All relevant data is within the manuscript and its supporting information files."

**Additional Editor Comments:**

The manuscript is interested meanwhile, the reviewers have raised a number of points which we believe would improve the manuscript and may allow a revised version to be published in PLOS one.

Reviewers' comments:

Reviewer's Responses to Questions

**Comments to the Author**

1. Is the manuscript technically sound, and do the data support the conclusions?

Reviewer #1: Yes

Reviewer #2: Yes

2. Has the statistical analysis been performed appropriately and rigorously? 

Reviewer #1: Yes

Reviewer #2: Yes

3. Have the authors made all data underlying the findings in their manuscript fully available?

Reviewer #1: Yes

Reviewer #2: Yes

4. Is the manuscript presented in an intelligible fashion and written in standard English?

Reviewer #1: Yes

Reviewer #2: No

5. Review Comments to the Author

Reviewer #1: Provide more details regarding the specific ethical considerations addressed in the study, such as informed consent procedures and data confidentiality measures. it would be beneficial to provide a succinct summary of the key findings before delving into the detailed interpretation. This will help readers grasp the main findings and their significance right at the beginning of the discussion.

Reviewer #2: Reviewing the paper titled “Recovery Time and Predictors of Severe Acute Malnutrition in Children Aged 6–59 Months via an Outpatient Therapeutic Program in Borena Zone: A Prospective Study”. The study contributes to the body of literature on the recovery time and predictors of SAM among children with unsevere conditions admitted to healthcare services.

Abstract: well written and provide a clear overview of the study. The conclusion should clearly mention that the results are only generalizable to children with SAM and to those who access healthcare services rather than every child with SAM. Those who seek health services have more severe symptoms.

Introduction: The study provides an overview of the literature, focusing on Ethiopia. Since Marasmus and edematous was mentioned later in the results, it would be good to define them and explain the difference.

Methods:

- Mention clearly what OTP intervention entails at the very beginning of the paper (types, doses etc)

- Precise whether the study sample have received a complementary treatment (other than OTP or from other services) to treat SAM. Mention if they were included in the sample or accounted for in the analysis

-The 117 inclusion criteria here is unclear: “Specifically, children who met the 117 inclusion criteria and were enrolled in a randomly chosen OTP service site during the 118 study period were considered as study participants”

- Is there a difference in the dose administered to children?

- How was the HHFD calculated? is it the sum of the consumption of each of the food groups? Precise how this has been calculated and if any weights have been used for highly nutritious food groups.

- For HHFD, how did you decide on the cutoff points for household food diversity score? Please add references or mention if the median or mean has been used

- Add a footnote on what does ‘other vegetables’ entail

Ethical consideration: provide further details on how children were matched at baseline and follow-up visits since the data was sent for analysis anonymously

Discussion: elaborate on how you addressed social desirability bias in the study, it is unclear.

Conclusion: This study must clearly state that the results are generalizable to children with uncomplicated severe acute malnutrition who accessed health facilities.

NB: Define the abbreviation when it appears for the first time in the text (e.g. HEW, SECA, etc).

6. PLOS authors have the option to publish the peer review history of their article (what does this mean?). If published, this will include your full peer review and any attached files.

Reviewer #1: No

Reviewer #2: No

---

## [Author Response · Author response to Decision Letter 0]

29 Jul 2024

III. Section 3: Review Comments to the Author

Reviewer#1: Comment to the authors and Responses given by the authors 

Please use the space provided to explain your answers to the questions above. You may also include additional comments for the author, including concerns about dual publication, research ethics, or publication ethics. (Please upload your review as an attachment if it exceeds 20,000 characters). 

Reviewer #1: 

1. Provide more details regarding the specific ethical considerations addressed in the study, such as informed consent procedures and data confidentiality measures. 

Response to Reviewer #1

1. With respect to the Ethical concerns raised

Dear Reviewer#1, We would like to express our sincere gratitude for your thorough review and valuable feedback regarding our manuscript. We appreciate the opportunity to clarify and address the concerns you raised, particularly concerning the ethical considerations associated with our study. 

We would like to reiterate that prior to initiating the study, we obtained ethical approval from the Institutional Review Board (IRB) at Dila University’s College of Health Sciences and Medicine. An official letter was then sent to the Borena zone health office, which further communicated to the relevant district health offices and outpatient therapeutic program (OTP) sites. Verbal consent was obtained from the mothers or caregivers before participation, ensuring that they fully understood the purpose of the study, the procedures involved, and their right to withdraw at any time. We emphasized the confidential nature of their responses and assured them that their participation would not impact their access to health services. 

To safeguard participant confidentiality, the data extraction forms did not include any identifying information of the children. Data was stored securely with limited access, employing password protection and strict data handling protocols. We used unique identification numbers linked to OTP registration to maintain data integrity while ensuring privacy. We took special care to inform caregivers clearly about the study's aims, which helped reduce any potential social desirability bias. This transparency was crucial in ensuring that caregivers provided accurate and honest information.

We have revised the manuscript to include these additional details, specifically in the methodology and ethical considerations sections (refer to the last part of the methods and material, in the Ethical Consideration sub-section on pages 12 for the updates). We believe these modifications address your concerns and strengthen the manuscript.

Thank you once again for your constructive feedback. We look forward to your further comments. Sincerely, Dr Girma Tenkolu, corresponding author.

2. It would be beneficial to provide a succinct summary of the key findings before delving into the detailed interpretation. This will help readers grasp the main findings and their significance right at the beginning of the discussion.

Response to Reviewer #1

Thank you, Reviewer #1, for your valuable feedback on our PLOS ONE manuscript. We appreciate your suggestion to include a succinct summary of the key findings at the beginning of the discussion, and we have revised this section accordingly.

To highlight the significance of our research immediately, we now open the discussion with a summary that includes: The study aimed to assess the median recovery time from severe acute malnutrition in 322 children in the Borena zone, Oromia region, Southern Ethiopia. Results indicated a recovery rate of 93.2%, demonstrating the effectiveness of the treatment protocols employed. The median recovery time was established at 42 days. Notably, the presence of symptoms such as diarrhea, vomiting, and edema at admission was identified as significant predictors that prolonged recovery time. Conversely, timely administration of routine medications, particularly Amoxicillin, was associated with faster recovery.

This adjustment aims to provide readers with a clear understanding of our key findings and their implications right from the outset. We believe this change will enhance the manuscript's clarity and impact.

Thank you once again for your constructive suggestions. We trust that these improvements address your concerns and strengthen our submission. For confirmation, please see the first paragraph of the discussion part of the revised manuscript starting from line No 1 up to the end. Sincerely, Dr Girma Tenkolu

Reviewer#2: Comment to the authors and Responses given by the authors 

Reviewer #2: 

1. Reviewing the paper titled “Recovery Time and Predictors of Severe Acute Malnutrition in Children Aged 6–59 Months via an Outpatient Therapeutic Program in Borena Zone: A Prospective Study”. The study contributes to the body of literature on the recovery time and predictors of SAM among children with unsevere conditions admitted to healthcare services.

Response Given

Dear Reviewer # 2 

 Thank you for your thoughtful and constructive feedback regarding our manuscript. We greatly appreciate the time and effort you have dedicated to reviewing our work, and we are committed to addressing your concerns thoroughly to enhance the quality of our manuscript.

We understand your observation regarding the study's contribution to the existing literature on severe acute malnutrition (SAM) and recovery time predictors. To clarify and amplify the relevance of our research, we will include a more comprehensive introduction that highlights the current state of knowledge on SAM recovery and emphasizes the importance of understanding recovery times and influencing factors, particularly in populations with varying admissions criteria. This will better contextualize our findings within the broader field of malnutrition research.

We will ensure that our results are communicated clearly, specifically detailing how the predictors identified (diarrhea, vomiting, edema, and timely medication) relate to recovery times. We believe that a more in-depth discussion of our findings will better illustrate the implications of our study for treatment strategies and public health policies in similar settings.

In response to your suggestion, we will include a section discussing potential directions for future research, such as exploring additional factors influencing recovery times or conducting similar studies in different regions or populations. This addition will provide valuable insights for researchers and practitioners aiming to address SAM more effectively.

We have revised the manuscript to incorporate these suggestions and improve clarity, coherence, and overall quality. We hope that these changes address your concerns and further establish the significance of our findings within the field.

Thank you once again for your valuable feedback. We look forward to your further comments and hope to fulfill the publication requirements of PLOS ONE. 

Sincerely, Dr Girma Tenkolu , Corresponding author

2. Abstract: well written and provide a clear overview of the study. The conclusion should clearly mention that the results are only generalizable to children with SAM and to those who access healthcare services rather than every child with SAM. Those who seek health services have more severe symptoms.

Response Given: Dear Reviewer # 2 

Thank you for your constructive feedback on our manuscript. We appreciate the time you took to review our work and your insightful comments.

We acknowledge your point regarding the generalizability of our findings. As per your suggestion, we have revised the conclusion to clarify that the results are specific to children experiencing severe acute malnutrition (SAM) who access healthcare services. In the revised manuscript, we emphasize that the outcomes might not be applicable to all children with SAM, particularly those who do not seek treatment or have less severe symptoms.

In the updated conclusion, we explicitly state: “These results are generalizable only to children with severe acute malnutrition who access healthcare services and may not reflect the broader population of all children with SAM, particularly those presenting with milder symptoms or those not seeking treatment.” This statement aims to ensure that readers appreciate the context and limitations of our findings.

We believe that this revision enhances the clarity and accuracy of our conclusions. Thank you once again for your invaluable feedback. We trust that these modifications align with your expectations and contribute to the overall quality of our manuscript. We look forward to your further comments. 

Sincerely, Dr Girma Tenkolu , Corresponding author

3. Introduction: The study provides an overview of the literature, focusing on Ethiopia. Since Marasmus and edematous was mentioned later in the results, it would be good to define them and explain the difference.

Here’s a revised version that includes definitions and explanations of marasmus and edematous malnutrition:

Response Given: Dear Reviewer # 2 

Thank you for your thoughtful feedback on our manuscript. We appreciate the time and effort you have dedicated to reviewing our work.

We acknowledge your suggestion regarding the inclusion of definitions and explanations for marasmus and edematous malnutrition in the Introduction section. To enhance the clarity of our manuscript and ensure that readers have a comprehensive understanding of these concepts, we have included the following definitions and distinctions: In the context of severe acute malnutrition (SAM), two primary forms are recognized: marasmus and kwashiorkor (edematous malnutrition). Marasmus is characterized by severe weight loss and muscle wasting due to inadequate energy intake, resulting in a noticeable loss of fat and muscle tissue. In contrast, kwashiorkor is primarily associated with protein deficiency and is often recognizable by the presence of edema, which causes swelling in the body, particularly in the abdomen and lower extremities. Understanding these distinctions is crucial for appropriate diagnosis and management of malnutrition in affected populations.

We believe that this addition enriches the manuscript by providing readers with essential background information that will enhance their understanding of the results presented later in the study.

Thank you again for your valuable insights. We trust that these changes address your concerns and improve the overall quality of our manuscript. We look forward to your further comments.

Sincerely, Sincerely, Dr Girma Tenkolu , Corresponding author

4. Methods:

i. Mention clearly what OTP intervention entails at the very beginning of the paper (types, doses etc)

Precise whether the study sample have received a complementary treatment (other than OTP or from other services) to treat SAM. Mention if they were included in the sample or accounted for in the analysis.

Response given : Dear Reviewer#2

We appreciate your valuable feedback and the opportunity to clarify the treatment regimen for the study sample. To address your inquiry, we confirm that all children included in our study exclusively received Ready-to-Use Therapeutic Food (RUTF) as part of the Outpatient Therapeutic Feeding Program (OTP). No complementary treatments were provided to any participants during the study period, nor were any children receiving additional therapies included in our analysis. This focus allowed us to assess the effectiveness of the OTP in treating Severe Acute Malnutrition (SAM) without the confounding effects of other interventions. We hope this response sufficiently addresses your concerns and clarifies the methodology of our study. Thank you for your constructive suggestions, which have helped enhance the clarity and rigor of our manuscript. Sincerely, Dr Girma Tenkolu

i. The 117 inclusion criteria here is unclear: “Specifically, children who met the 117 inclusion criteria and were enrolled in a randomly chosen OTP service site during the 118 study period were considered as study participants”

Response given : Dear Reviewer#2

Thank you for your valuable feedback regarding our manuscript submitted to PLOS ONE. We appreciate the reviewers’ insights, which have helped us enhance the clarity and rigor of our study. 

In response to the concern about the clarity of our inclusion criteria, we acknowledge that the phrasing may have caused some confusion. To clarify, children eligible for inclusion in our study were defined by the following criteria:

• Participants must be between 6 to 59 months old.

• All children must have been diagnosed with uncomplicated severe acute malnutrition as per the World Health Organization guidelines.

• Children must have passed an appetite test prior to enrollment.

• Eligible participants displayed bilateral pitting edema and a Weight-for-Length/Height (WFL/WFH) measurement of less than -3 Z-scores, indicating the severity of malnutrition without any accompanying medical complications.

• Children were enrolled from randomly selected Outpatient Treatment Program (OTP) service sites during the study period, which lasted from March 1 to March 30, 2023.

We have amended the manuscript to include a more detailed description of the inclusion criteria, making it explicit for readers. This addition should enhance the transparency of our methodology and allow for a better understanding of the study population.

Thank you again for your constructive feedback. We are committed to addressing all reviewer concerns and improving our manuscript for publication. For more confirmation, please refer the method section on page 7, in the sub-section of Study population and Inclusion and Exclusion Criteria, line No 1 to end. Sincerely, Dr Girma Tenkolu

ii. Is there a difference in the dose administered to children?

Response given 

Dear Reviewer#2, Thank you for your insightful feedback regarding our manuscript submitted to PLOS ONE. We appreciate your attention to detail and your valuable suggestions for improving our work.

In response to the question regarding differences in the doses administered to children, we would like to clarify the following: 

• The treatment protocols followed in our study adhered to the standardized guidelines established by the World Health Organization for the management of severe acute malnutrition in children. The dosage of Amoxicillin, specifically, was determined based on the weight of the children and followed the recommended guidelines for treatment of infections common in this population. 

• The doses were carefully calculated for each child based on their weight and clinical condition at the time of treatment. For instance, children typically received 20-40 mg/kg/day of Amoxicillin divided into two doses. This approach ensures that each child receives an appropriate dosage tailored to their specific needs. 

• We maintained detailed records of the medication administered, including dosages, to ensure compliance with treatment protocols and the ability to assess any impact on recovery times.

To enhance clarity in our manuscript, we will include additional details regarding the dosing protocols for Amoxicillin and other medications used in our study. We believe this will address your concerns and provide a clearer understanding of our methodology. Thank you once again for your constructive comments. We are dedicated to refining our manuscript in response to all reviewer feedback and ensuring it meets the publication standards. For more confirmation, please refer the method section on page 12-13, in the sub-section of Independent variables:, line No 15 to end. Sincerely, Dr Girma Tenkolu

iii. How was the HHFD calculated? is it the sum of the consumption of each of the food groups? Precise how this has been calculated and if any weights have been used for highly nutritious food groups. For HHFD, how did you decide on the cutoff points for household food diversity score? Please add references or mention if the median or mean has been used

Response given : Dear Reviewer#2

Thank you for your insightful questions regarding the calculation of the Household Food Diversity Score (HHFD) within our study. We appreciate your attention to detail, as it aids in enhancing the clarity of our methodology.

The HHFD was assessed using a standardized tool that evalu

---

## [Decision Letter · Decision Letter 1]

18 Sep 2024

PONE-D-24-07649R1Recovery Time and Predictors of Severe Acute Malnutrition in Children Aged 6–59 Months via an Outpatient Therapeutic Program in Borena Zone: A Prospective Study.PLOS ONE

Dear Dr. Bune,

Thank you for submitting your manuscript to PLOS ONE. After careful consideration, we feel that it has merit but does not fully meet PLOS ONE’s publication criteria as it currently stands. Therefore, we invite you to submit a revised version of the manuscript that addresses the points raised during the review process.

We look forward to receiving your revised manuscript.

Kind regards,

Ammal Mokhtar Metwally, Ph.D (MD)

Academic Editor

PLOS ONE

Journal Requirements:

Additional Editor Comments:

The manuscript is interested meanwhile, the reviewers have raised a number of points which we believe would improve the manuscript and may allow a revised version to be published in PLOS one.

Reviewers' comments:

Reviewer's Responses to Questions

**Comments to the Author**

1. If the authors have adequately addressed your comments raised in a previous round of review and you feel that this manuscript is now acceptable for publication, you may indicate that here to bypass the “Comments to the Author” section, enter your conflict of interest statement in the “Confidential to Editor” section, and submit your "Accept" recommendation.

Reviewer #2: All comments have been addressed

2. Is the manuscript technically sound, and do the data support the conclusions?

Reviewer #2: Yes

3. Has the statistical analysis been performed appropriately and rigorously? 

Reviewer #2: Yes

4. Have the authors made all data underlying the findings in their manuscript fully available?

Reviewer #2: Yes

5. Is the manuscript presented in an intelligible fashion and written in standard English?

Reviewer #2: Yes

6. Review Comments to the Author

Reviewer #2: Thank you to the authors for taking the time to respond to the comments point-by-point. The manuscript now looks great and addresses all the concerns.

I have one minor comment: Could you please specify why 4.5 was used as a cutoff point for HHFS? Was this choice based on literature, or did it align with the mean/median in your data? if it is based on the literature, provide references.

Best of luck with the publication of your paper.

7. PLOS authors have the option to publish the peer review history of their article (what does this mean?). If published, this will include your full peer review and any attached files.

Reviewer #2: No

---

## [Author Response · Author response to Decision Letter 1]

3 Oct 2024

Reviewers' comments:

Reviewer's Responses to Questions

Comments to the Author

1. If the authors have adequately addressed your comments raised in a previous round of review and you feel that this manuscript is now acceptable for publication, you may indicate that here to bypass the “Comments to the Author” section, enter your conflict of interest statement in the “Confidential to Editor” section, and submit your "Accept" recommendation.

Reviewer #2: All comments have been addressed

2. Is the manuscript technically sound, and do the data support the conclusions?

Reviewer #2: Yes 

3. Has the statistical analysis been performed appropriately and rigorously?

Reviewer #2: Yes

4. Have the authors made all data underlying the findings in their manuscript fully available?

Reviewer #2: Yes 

5. Is the manuscript presented in an intelligible fashion and written in standard English?

Reviewer #2: Yes 

6. Review Comments to the Author

Reviewer #2: Thank you to the authors for taking the time to respond to the comments point-by-point. The manuscript now looks great and addresses all the concerns.

I have one minor comment: Could you please specify why 4.5 was used as a cutoff point for HHFS? Was this choice based on literature, or did it align with the mean/median in your data? if it is based on the literature, provide references.

Best of luck with the publication of your paper. 

7. PLOS authors have the option to publish the peer review history of their article (what does this mean?). If published, this will include your full peer review and any attached files.

Do you want your identity to be public for this peer review? For information about this choice, including consent withdrawal, please see our Privacy Policy.

Reviewer #2: No

Here’s a point-by-point response letter addressing Reviewer #2's comments on your manuscript. This format will help you clearly communicate how you have addressed their question and comments. 

Responses given to the Review Comments 

Responses Review Comments to the Author# 6 to the specific minor comment: Could you please specify why 4.5 was used as a cutoff point for HHFS? Was this choice based on literature, or did it align with the mean/median in your data? if it is based on the literature, provide references.

Dear Reviewer #2,

Thank you for your thoughtful review and for acknowledging the improvements made to our manuscript. We appreciate your feedback and have addressed your comment regarding the cutoff point for the Household Food Security (HHFS) measure. 

We are grateful for your request for clarification on the use of 4.5 as a cutoff point for the HHFS. In the revised manuscript, we have provided a detailed explanation in the Methods section. This cutoff was selected based on its alignment with existing literature, which indicates that this threshold effectively distinguishes between food secure and food insecure households (relevant references are now included in the Methods and Materials section, on the sub topic of Operational definition section, Page 12-14, Paragraph 2, starting line Nu 8). Additionally, this cutoff closely corresponds to the median score observed in our data, ensuring that our analysis is pertinent to our population while maintaining consistency with established research in this field.

We appreciate your careful consideration of our work and your encouragement regarding the publication of our paper. We believe these additions enhance the clarity and rigor of our manuscript.

Thank you once again for your constructive feedback.

Kind regards, 

Dr. Girma Tenkolu Bune, Corresponding Author 

Dilla University

---

## [Decision Letter · Decision Letter 2]

21 Oct 2024

Recovery Time and Predictors of Severe Acute Malnutrition in Children Aged 6–59 Months via an Outpatient Therapeutic Program in Borena Zone: A Prospective Study.

PONE-D-24-07649R2

Dear Dr. Bune,

We’re pleased to inform you that your manuscript has been judged scientifically suitable for publication and will be formally accepted for publication once it meets all outstanding technical requirements.

Kind regards,

Ammal Mokhtar Metwally, Ph.D (MD)

Academic Editor

PLOS ONE

Additional Editor Comments (optional):

Thank you for addressing all reviewers' comments. We believe your manuscript is ready for publication

Reviewers' comments:

Reviewer's Responses to Questions

**Comments to the Author**

1. If the authors have adequately addressed your comments raised in a previous round of review and you feel that this manuscript is now acceptable for publication, you may indicate that here to bypass the “Comments to the Author” section, enter your conflict of interest statement in the “Confidential to Editor” section, and submit your "Accept" recommendation.

Reviewer #2: All comments have been addressed

2. Is the manuscript technically sound, and do the data support the conclusions?

Reviewer #2: Yes

3. Has the statistical analysis been performed appropriately and rigorously? 

Reviewer #2: Yes

4. Have the authors made all data underlying the findings in their manuscript fully available?

Reviewer #2: Yes

5. Is the manuscript presented in an intelligible fashion and written in standard English?

Reviewer #2: Yes

6. Review Comments to the Author

Reviewer #2: The authors responded to my comments in revision 1 and 2. I have no further comments. Congratulations on your paper!

7. PLOS authors have the option to publish the peer review history of their article (what does this mean?). If published, this will include your full peer review and any attached files.

Reviewer #2: No

---

## [Editor Report · Acceptance letter]

29 Oct 2024

PONE-D-24-07649R2 

PLOS ONE

Dear Dr. Bune, 

I'm pleased to inform you that your manuscript has been deemed suitable for publication in PLOS ONE. Congratulations! Your manuscript is now being handed over to our production team.

Kind regards, 

on behalf of

Professor Ammal Mokhtar Metwally 

Academic Editor

PLOS ONE